# Particulate Air Pollution and Risk of Neuropsychiatric Outcomes. What We Breathe, Swallow, and Put on Our Skin Matters

**DOI:** 10.3390/ijerph182111568

**Published:** 2021-11-03

**Authors:** Lilian Calderón-Garcidueñas, Elijah W. Stommel, Ravi Philip Rajkumar, Partha S. Mukherjee, Alberto Ayala

**Affiliations:** 1College of Health, The University of Montana, Missoula, MT 59812, USA; lilian.calderon-garciduenas@umontana.edu; 2Universidad del Valle de México, Mexico City 14370, Mexico; 3Department of Neurology, Geisel School of Medicine at Dartmouth, Hanover, NH 03755, USA; Elijah.W.Stommel@hitchcock.org; 4Department of Psychiatry, Jawaharlal Institute of Postgraduate Medical Education and Research, Pondicherry 605006, India; ravi.psych@gmail.com; 5Interdisciplinary Statistical Research Unit, Indian Statistical Institute, Kolkata 700108, India; psmukherjee.statistics@gmail.com; 6Sacramento Metropolitan Air Quality Management District, Sacramento, CA 95814, USA; 7Department of Mechanical and Aerospace Engineering, West Virginia University, Morgantown, WV 26506, USA

**Keywords:** Alzheimer’s, Parkinson’s, quadruple aberrant proteins, particulate matter, air pollution, nanoparticles, pollution neurology and psychiatric outcomes, ultrafine particles, internal combustion emissions, wildfires

## Abstract

We appraise newly accumulated evidence of the impact of particle pollution on the brain, the portals of entry, the neural damage mechanisms, and ultimately the neurological and psychiatric outcomes statistically associated with exposures. PM pollution comes from natural and anthropogenic sources such as fossil fuel combustion, engineered nanoparticles (NP ≤ 100 nm), wildfires, and wood burning. We are all constantly exposed during normal daily activities to some level of particle pollution of various sizes—PM_2.5_ (≤2.5 µm), ultrafine PM (UFP ≤ 100 nm), or NPs. Inhalation, ingestion, and dermal absorption are key portals of entry. Selected literature provides context for the US Environmental Protection Agency (US EPA) ambient air quality standards, the conclusions of an Independent Particulate Matter Review Panel, the importance of internal combustion emissions, and evidence suggesting UFPs/NPs cross biological barriers and reach the brain. NPs produce oxidative stress and neuroinflammation, neurovascular unit, mitochondrial, endoplasmic reticulum and DNA damage, protein aggregation and misfolding, and other effects. Exposure to ambient PM_2.5_ concentrations at or below current US standards can increase the risk for TIAs, ischemic and hemorrhagic stroke, cognitive deficits, dementia, and Alzheimer’s and Parkinson’s diseases. Residing in a highly polluted megacity is associated with Alzheimer neuropathology hallmarks in 99.5% of residents between 11 months and ≤40 y. PD risk and aggravation are linked to air pollution and exposure to diesel exhaust increases ALS risk. Overall, the literature supports that particle pollution contributes to targeted neurological and psychiatric outcomes and highlights the complexity of the pathophysiologic mechanisms and the marked differences in pollution profiles inducing neural damage. Factors such as emission source intensity, genetics, nutrition, comorbidities, and others also play a role. PM_2.5_ is a threat for neurological and psychiatric diseases. Thus, future research should address specifically the potential role of UFPs/NPs in inducing neural damage.

## 1. Introduction

We live surrounded by environmental hazards including toxic compounds in air, water, soil, and food. Pollution is ubiquitous and mostly unavoidable. Neurotoxic organic and inorganic compounds come from fossil fuel combustion, engineered nanoparticles, nanoplastics, and compounds resulting from disasters such as forest wildfires. We are constantly exposed to these environmental hazards, regardless of age, sex, or socioeconomic status. Some people are more vulnerable than others and the brain in development is a target. The complexities of our modern-day, fossil fuel-based society are a factor when we talk about central nervous system (CNS) effects and the role of specific pollutants.

Minding only criteria pollutants [1] does not give us the full picture of exposure risk because these six compounds are regulated primarily for meeting regional air quality standards, rather than explicitly for reducing chronic or acute exposures and causality of adverse health outcomes. Other policies and standards exist for limiting pollutant-specific near-source exposures. Furthermore, criteria pollutants measured in ambient air (ozone (O_3_), particulate matter (PM), sulfur dioxide (SO_2_), nitrogen dioxide (NO_2_), carbon monoxide (CO), and lead (Pb)) are operationally defined. Comparisons of pollution burdens across regions and countries must begin by ensuring the application of identical measurement methods and analytical techniques, otherwise we risk comparing apples to oranges. In addition, PM and O_3_ are not singular entities; rather, they are lumped parameters representing a family of source-dependent constituents and precursors that can vary greatly in space and time and in physicochemical and toxicological characteristics. For these reasons, it is extremely difficult to make comparisons of PM_2.5_ pollution and health effects in, for example, Delhi versus New York City or Provo, Utah versus Metropolitan Mexico City. Moreover, the determinants of adverse health outcomes are numerous: the strength and nature of emission sources, exposure times, and cumulative exposures over a lifetime, age, morbidities, and occupational history are some of the most obvious. The cumulative effect of these and other factors will determine specific health outcomes and responses vary significantly by population in terms of genetics, nutrition, exercise patterns, and cultural factors. These arguments should be an alarm for the reader. The sophistication of instruments, methodologies, and models to characterize air pollutants, exposures, or the results and interpretation of a specific PM_2.5_ study, for example, may not equate to ability to understand and compare results, not even within the same country, let alone across the globe. Thus, local capacity to understand emission sources, implement PM control measures, and establish clinical, laboratory, and pathology links becomes paramount. 

There is strong evidence of causality between PM_2.5_ air pollution exposure and cardiovascular morbidity and mortality [2]. The 2021 WHO Global Air Quality Guidelines recommendation for annual PM_2.5_ Air Quality Guideline Level is 5 µg/m^3^, with four interim targets proposed as incremental steps in a progressive reduction of air pollution and intended for use in highly polluted areas [3]. General practitioners, emergency room doctors, internists, neurologists, psychiatrists, pediatricians, cardiologists, infectologists (think COVID), and others see firsthand the morbidity and mortality effects of air pollution and need readily available information about how to interpret results across locations.

This Review provides information about PM CNS effects with a focus on stroke, neurodegenerative diseases, and psychiatric symptoms preceding or concomitant with neurodegeneration. Our discussion covers the current policy landscape in the US regarding particle pollution, internal combustion engine PM emissions, the differences in ambient air quality standards between the US and WHO 2021 guidelines, the recent expert suggestions of an Independent Particulate Matter Review Panel, and the importance of UFPs and NPs for the mechanistic neural damaging pathways linked to exposure to them. Additionally, the reader will find arguments of specific neural damage from PM exposures and related NP experimental work, epidemiological studies establishing robust links between PM and neurological and psychiatric adverse outcomes, our opinion of the role of health workers in protecting patients from PM pollution, and lessons to be learned.

## 2. Particulate Matter, US Air Pollution Standards, Ultrafine Particles, Engineered Nanoparticles, and Internal Combustion Engine Emissions

Last year, the US EPA decided to retain, without revision, the existing primary and secondary National Ambient Air Quality Standards (NAAQS)^1^ for fine particulate matter PM_2.5_ (particles ≤ 2.5 µm). The primary standards are the focus of our concern in terms of potential adverse effects on the brain. The primary annual average and 24 h standards are 12.0 µg/m^3^ and 35.0 µg/m^3^, respectively [1]. These pollution caps are designed to protect public health from the effects associated with both long- and short-term exposures. However, because many areas of the country are not in attainment of these standards, excess pollution in the air is responsible for thousands of premature deaths in the US [4]. The suggestions of an expert Independent Particulate Matter Review Panel [4] included: “*an annual standard between 10 μg per cubic meter and 8 μg per cubic meter would protect the general public and at-risk groups”* and “*the 24-h standard be set between 30 μg per cubic meter and 25 μg per cubic meter*”. At the time of this writing, the EPA is revisiting its decision and undertaking another assessment. Keeping these observations, the 2021 WHO Global Air Quality Guidelines [3], and the Panel suggestions in mind, many of the neurological and psychiatric manifestations associated with PM_2.5_ are indeed related to exposures below the current US EPA standards [5]. 

The anthropogenic sources of PM_2.5_ pollution across the world include industrial fuel combustion, wood burning, farming operations, construction and demolition, dust, managed burning, wildfires, and internal combustion engines powering vehicles, ships, locomotives, and equipment. Motor vehicle brake and tire wear are a growing PM_2.5_ source. Combustion of any fossil fuel or biofuel generates PM_2.5_ emissions and diesel engines are a particularly notorious source. The US has adopted aggressive emission standards for heavy-duty engines. Since 2007, the limit is no more than 10 milligrams of PM mass emissions per brake-horsepower-hour of energy output (10 mg/bhp-hr). However, this standard does not explicitly limit the emissions in the UFP size range [6]. Europe has moved to regulate some UFP emissions by adopting an explicit limit on the number of solid particles emitted by an engine. In contrast, it is unlikely the US will ever adopt a similar standard [7]. 

The contributions from tailpipe and non-tailpipe traffic sources to the traffic-related air pollution (TRAP) in the near-roadway environment constitute a key source of small size PM: 66%, 32%, and 18% of PM_0.2_, PM_2.5_, and PM_2.5–10_ mass, respectively [8]. Thus, given the slow transition towards electric, zero tailpipe emission mobility, there will still be significant human exposure to traffic-related combustion and non-tailpipe vehicle emissions for decades. Moreover, brake and tire wear, catalyst degradation, and resuspended road dust in the quasi-ultrafine size range (PM_0.2_), will be increasingly important sources of PM_0.2_ and metals in the future [8]. Climate change is increasing the frequency and severity of wildfires globally contributing extraordinary amounts of PM_2.5_ pollution and significant exposures for thousands of firefighters and millions of people in the smoke path. More than 50 million homes are located in the land/urban interface (WUI) in the US and projections exist of an increment of 1 million houses in the WUI every 3 y [9]. Burke and coworkers [9] used a statistical model that linked satellite-based fire and smoke data to information from pollution outdoor monitoring stations and estimated that wildfires have accounted for up to 25% of PM_2.5_ in recent years across the US, and up to 50% in certain Western regions. Of utmost importance for this Review, wildfire PM_2.5_ concentrations are on average 150.0 μg/m^3^ with recorded concentrations above 500 μg/m^3^ and with spatial patterns that do not follow traditional socioeconomic pollution exposure gradients [9,10]. These are key pieces of information to keep in mind for future data collection on incidence and prevalence of major neurological and psychiatric diseases in the most affected regions (Figure 1). 

All internal combustion engines produce UFP emissions irrespective of the fuel or lubricant used. These can be volatile, semi-volatile, or solid particles. While UFPs are not regulated explicitly in the US, extensive research over the last two decades has led to a growing understanding of the sources and precursors, the mechanisms of formation in engine exhaust and in the atmosphere, metrology, and emission control approaches. However, clearly, some knowledge gaps remain. One important lesson learned is that limits on PM_2.5_ mass emissions may or may not limit UFP emissions. The correlation between PM and UFPs or NPs depends on several factors and, in some cases, PM and UFP emissions are inversely related. The mass of PM emissions is a conserved quantity. The number of particles in those emissions is not. In addition, the US has no NAAQS for UFPs, although the definitions of PM_2.5_ and PM_10_ encompass these particles. PM_2.5_ ambient concentrations tend to be spatially homogeneous and regionally distributed. In contrast, UFP number concentrations in the air exhibit sharp gradients as a function of distance from a source. The atmospheric processes that lead to formation of particles from precursors are not well understood. The lack of UFP NAAQS has also discouraged the creation of a permanent network of ambient monitoring stations for UFP concentrations. Such a network would facilitate long-term epidemiological studies. Instead, exposure assessment of UFPs has to rely on other methods such as use of regression models or personal monitoring. For these reasons, much remains to be discovered about the explicit role that human exposure to UFP pollution plays in the well-documented morbidity and mortality effects of exposure to PM_2.5_.

## 3. Engineered Nanoparticles

Engineered nanoparticles are also in our environment, thus we inhale, swallow, and apply them to our skin regularly. Household and commercial products with TiO_2_ or SiO_2_ in pigments and foods are significant sources of NPs released to the environment [11]. According to Zheng and Nowack [11] back in 2016, 50% of the nanosize TiO_2_ particles released into wastewater came from the environmental release of 22,400 tons of TiO_2_ particles in pigments. Solid inorganic NPs (TiO_2_ and ZnO) are used as UV filters in sunscreen formulations, causing serious concerns over the efficiency and safety of existing chemical and physical UV filters used in consumer products [12]. Iron oxide magnetic, combustion-derived UFPs pose added cellular damage mechanisms: magnetic NPs can be heated by oscillating magnetic fields and produce local hyperthermia, damage cellular organelles, and increase oxidative stress [13]. 

## 4. Portals of Entry to the Brain and Key Neural Damage Mechanisms

The respiratory tract is a key portal of entry through inhalation as the air passes through the nasal cavity, directly to the olfactory nerve and olfactory bulb, the trigeminal, facial, and glossopharyngeal nerves, and to the lower respiratory tract and the alveoli [14]. UFPs accumulated in children’s nasal epithelium, the olfactory bulb, and frontal cortex and extensive damage of the neurovascular unit (NVU) are documented in highly exposed Metropolitan Mexico City (MMC) young residents: capillary leakage, vascular amyloid, amyloid plaques, and hyperphosphorylated tau (P-tau) accumulation are present in the first two decades of life [15]. The gastrointestinal (GI) system is an important portal through ingestion: we swallow a significant amount of the inhaled UFPs and ingest NPs added to food ingredients, supplements, food and drink containers, toothbrushes, and toothpaste. Titanium dioxide and silicon dioxide NPs are widely used as additives in the food industry and their long-time dietary intake produces gut microbiota disruption, intestinal, kidney, spleen, and liver injury, and neurotoxic effects through gut–brain axis disruption, with a major impact on children [16]. Swallowed NPs/UFPs have easy access to the GI epithelium and submucosa and their damage allows direct access to the enteric nervous system [17]. In a combined study of gastric, small bowel, and vagal nerves at the cervical level in children, young adults, and dogs with low versus high exposures to PM_2.5_, NPs were abundant in erythrocytes, unmyelinated submucosal, perivascular, and intramuscular nerve fibers, ganglionic neurons, and vagal nerves and associated with organelle pathology, in highly exposed urbanites [17]. Immunohistochemistry showed hallmarks of Parkinson’s and Alzheimer’s diseases including aggregated alpha-synuclein and P-tau in gastrointestinal tract and vagal nerves of young children and Mexico City young adults [17]. The skin and mucosae are also critical portals of entry especially in certain occupational environments (fire—nanoparticle extinguishing agents), and in people using UV screen protectors [12]. 

Once NPs reach the systemic circulation, they can travel freely or hitchhike on red or white blood cells and start their journey through the body [14,18]. The NVU [19] is an early NP target and since the blood–brain barrier (BBB) breakdown contributes to blood barrier dysfunction, vascular leaks, and associated cognitive decline, the extensive NVU damage associated with NPs and documented in urban toddlers is likely an early step in the brain damage cascade [14,15]. Damage to the NVU complex functional and anatomical structure impacts endothelium, pericytes, astrocytes, microglia, capillaries, arterioles, basal lamina covered by pericytes, and neural cells including neurons, interneurons, and extracellular matrix [19]. NVU cells and matrixes interact closely for maintaining homeostasis of neurovascular coupling, BBB integrity, and trans-endothelial fluid transport. Any disruption of the NVU will translate into failure of tight coupling between neural activity and cerebral blood flow (CBF) [19]. Damage to red blood cells (RBCs) and brain endothelium associated with NPs [14] is clearly contributing to vasopathological effects and compromising the interaction between RBCs, endothelial cells, platelets, and macrophages, and ultimately, as Petrini and coworkers [20] discussed, altering thrombosis, hemostasis, and immune responses. NVU alterations predict brain dysfunction, neuroinflammatory responses, and neurodegeneration. 

Dysfunctional mitochondria are an early feature of neurodegenerative diseases [21] and early responses to excessive production of reactive oxygen species (ROS) result in the induction of mitophagy to remove damaged mitochondria. Mitochondrial damage associated with sustained oxidative stress, as in the exposure to air pollutants and NPs, overwhelms autophagic and mitophagic pathways, activating a vicious circle that leads to reduced capacity to remove damaged mitochondria, and/or an alteration in the regulation of mitophagy [21]. Defective mitophagy leads to synaptic degeneration and mitochondrial fragmentation in association with Aβ and P-tau and cognitive dysfunction in Alzheimer’s disease (AD) [21]. NPs are very effective in targeting the cell powerhouse in neural cells in air pollution-exposed children and young adults (Figure 2). 

A major pathway involved in NPs’ detrimental brain effects is ROS production [22]. The oxidative reactivity of NPs—over other physicochemical properties—causes cytotoxicity via induction of cellular oxidative stress, and a good example is combustion-associated Fe oxide NPs, both magnetite and maghemite, which once in a cell’s acidic lysosomal environment release free Fe ions [22]. Inflammatory responses are followed by anti-inflammatory actions, in response to metal base NPs, including superparamagnetic Fe oxide NPs, and autophagy and impaired neovascularization are serious and unwanted neural responses [22]. The size of NPs and chemical composition are critical for their blood circulation time, blood vessel damage, and neural toxicity [22]. Gutiérrez et al. [23] used 14 and 22 nm magnetic NPs with the same core but subjected to different surface modifications procedures; the results were NPs with different size aggregates and arrangements impacting their magnetic and heating properties. Gutiérrez and coworkers’ observations [23] are critical for environmental magnetic NPs and brain effects: (1) colloidal stability depends on the balance of magnetic, dipolar, and van der Waals forces and repulsive interactions, mainly electrostatic and steric, and (2) when magnetic cores are assembled, the distance between cores affects magnetic behavior of the entire particle, i.e., dipole–dipole interactions result in an alteration of magnetic properties. In a live organism where NPs can be highly concentrated within endosomes, Gutiérrez et al. suggested magnetic interactions become very strong and exposure to alternating magnetic fields (AMFs) gives rise to heat produced by NPs and mainly determined by size, shape, crystal structure, saturation magnetization, and susceptibility [23]. This information is very relevant to urban dwellers with significant concentrations of magnetic NPs in their brains [14].

All NPs in biological fluids are surrounded by a corona and protein corona–NP binding affinity depends on shape, size, and surface characteristics of NPs and hydrodynamic, electrodynamic, and magnetic forces [22,23,24]. High-affinity proteins forming coronas are stable, and protein adsorption on the surface of NPs will allow the protein to keep most of their native conformation, but as a result, the protein could be thermodynamically unstable. The protein adsorption depends on the available protein concentration and in environments with several proteins, the NP surface may bind several of them [24]. It is critical to remember that the interaction of NPs with organelles occurs mostly through the NP–corona [24], i.e., the same NP coated with different proteins will have easier access to a cell if the adsorbed protein unfolding helps in accessing cell surface receptors or inhibited if the protein structure is lost. As a result, the protein corona, the interaction of each individual protein, and the NP size influence larger surface concentration, cellular uptake, accumulation, degradation, and clearance of the NP. NPs can act as molecular chaperones to carry out protein folding, destabilization, and protein aggregates and as a result of elevation of protein concentrations and destabilization of protein, folded states promote amyloid aggregates [24,25]. 

The molecular chaperone function and the cellular efficiency of the degradation systems are key for the correct folding of the cellular protein and for preventing aggregation. Certain surfaces are very effective in promoting amyloid formation (i.e., those with lipid bilayers, collagen fibers, and polysaccharides). Hartl’s work [25] emphasized that misfolded proteins expose hydrophobic amino acid residues and regions of unstructured polypeptide backbone, resulting in cytosolic protein aggregates rich in β-sheet structures, i.e., α-synuclein and tau. The common denominator of amyloid-prone proteins is their small size and the fact that are intrinsically disordered in non-aggregated states [25]. 

We also need an intact endoplasmic reticulum (ER) for eliminating abnormal proteins, thus any ER alterations will result in a protein degradation failure. Highly exposed urban megacity residents have severe alterations in the ER, widespread degenerated mitochondria–ER contacts (MERCs), and quadruple protein misfolding in close association with NPs [26].

In the scenario of PM_2.5_ pollution and NPs, the current literature supports the following: (1) environmental NPs can directly alter brain proteins [14,15,16,17,22,23,24]; (2) since NPs can travel anterograde, retrograde, and trans-synaptically from any portal of entry, altered proteins are capable of moving in all directions [14,15,17]; (3) protein misfolding, aggregation, and fibrillation take place in neural cells and follow the NP path’s initial portal of entry (i.e., brainstem) [17,22,23,24]; and (4) neuroinflammation, damage to the neurovascular unit, oxidative stress, and magnetic effects all potentially contribute to neural damage [22,23,24,25,26]. NPs can potentially elicit key mechanistic pathways as described in all major neurodegenerative diseases and produce the neuropathological features of several diseases at the same time, as we are observing in Mexico City children and young adults (Figure 3) [14,15,17,26].

## 5. Stroke and Air Pollution

We have had strong evidence in the last two decades of the causal role of PM_2.5_ air pollution and cardiovascular morbidity and mortality, including stroke [2,3,27]. The incidence of stroke increases with age, but as many as 20% of cases are seen in young adults [28] and air pollution is a risk factor in ischemic stroke, affecting highly air pollution-exposed subjects, including those with a history of occupations with high PM_2.5_ exposures. Given that the stroke risk associated with air pollution is 29% globally [27], and the higher risk is present in low- and middle-income countries, neurologists in high-risk regions and the ones practicing in highly polluted US cities with minority groups (i.e., African Americans and Hispanics) need to be aware of the data [28,29]. Factors such as inflammation, oxidative stress, and endothelial dysfunction associated with ambient air pollution exposures are targeting cerebrovascular and neuropsychiatric disorders [30]. Transient ischemic attacks (TIAs) are also reported along with cerebrovascular disease and ischemic stroke associated with PM_2.5_ increments of 10 μg/m^3^ in same-day hospital admissions [31]. Remarkably, even short-term air pollution winter exposures in a relatively clean environment were associated with hospitalization for all strokes in a large urban center in Ireland [32]. Black carbon and PM_10_ from road wear, traffic exhaust, and residential heating were associated with stroke and ischemic heart disease in three Swedish cities (Gothenburg, Stockholm, and Umeå), with relatively low PM_2.5_ levels (i.e., average 9.2 µg/m^3^, range of 2.9 to 17 µg/m^3^); a piece of information very relevant for vulnerable populations [33]. Thus, stroke risk is high in areas where PM_2.5_ ambient concentrations are above the US EPA standard [28,29], but as the Ljungman paper [33] shows, low PM from local traffic exhaust sources also carries a risk for stroke.

**Core message:** Modifiable risk factors are 90% responsible of the stroke burden across the world, including indoor and outdoor air pollution and lead exposure. Stroke is a leading cause of morbidity and mortality in the United States and young adults are at risk.

## 6. Neurodegenerative Diseases and Nanoparticles

Epidemiological studies across countries support strong associations between risk of cognitive deficits, dementia, and PM exposures, including long-term PM_2.5_ exposures below existing US EPA standard [5,34,35,36,37]. The critical amount or time of UFP/NP exposure needed for neurodegeneration development under urban exposure scenarios is not clear, however, for MMC residents an eleven-month-old baby already showing brainstem hyperphosphorylated tau had 20 μg/m^3^ cumulative PM_2.5_ (CPM_2.5_; calculated for age at death + pregnancy time) [15,26]. This level of exposure is much lower compared to the 2522 μg/m^3^ calculated for a 39-year-old with AD neurofibrillary tangle advanced stages V–VI and suggests the neurodegenerative process is at work in early pediatric ages with short exposure times, including intrauterine life. Elderly populations with cerebrovascular disease show accelerated cognitive decline associated with PM pollution [37] and, strikingly, higher risk of dementia is associated with living less than 50 m from a major traffic road, predominantly affecting residents in major cities and people who never moved from their urban location [35]. The issue of heavy traffic and exposures above the PM_2.5_ US EPA standard carrying higher dementia risk should be kept in mind for large city dwellers, highly industrial polluted towns—regardless of size—and for occupational exposures (i.e., taxi drivers) [38].

Parkinson’s disease risk, aggravation of the disease, and increased risk of first hospital admission have been linked to air pollution in the USA and elsewhere [39,40,41]. It is also important to remember that environmental airborne manganese present in PM_2.5_ impacts millions of people and is associated with Mn found in fuel combustion, soil contamination in mining settings, welder workers’ exposures, and industrial smelting and steelmaking [42,43]. Residents in proximity to an industrial area can have exposures of PM_2.5_–Mn concentrations as high as 203 ng/m^3^ that could cause clinical parkinsonism [43].

Amyotrophic lateral sclerosis (ALS) is no exception in the literature linking the risk to PM pollution, metal, and metalloid exposures, diesel exhaust, and specific occupations [44,45,46,47] and efforts are in place to recruit population-based controls for an epidemiologic case–control study to examine ALS environmental risk factors [48]. Andrew and coworkers [47] examined 188 ALS patients from New England and Ohio and reported that lead is strongly associated with ALS risk (adjusted OR 2.92, 95% CI 1.45–5.91), along with severe electrical burns and head trauma (OR 2.86, 95% CI 1.37–6.03 and OR 1.60 95% confidence interval (CI) 1.04–2.45, respectively). In the US, ALS affects predominantly non-Hispanic white males, ≥60 years, and those with a family history of ALS [44]. An increasing incidence and prevalence of ALS across the world and the increased risk of ALS in association with diesel exhaust and specific occupations (i.e., mechanics, painting, construction, airport workers, and jet pilots) are of concern for the general public and also regarding general military service and/or specific exposures associated with military service measured among military personnel [49,50]. Although ALS and frontotemporal dementia (FTD) are both linked to TDP-43 nuclear depletion and concurrent cytoplasmic accumulation in vulnerable neurons, there are no works directly linking air pollution with impact on FTD risk, except Numan et al. [51], addressing oxidative stress and autophagy in FTD and other neurodegenerative diseases, and the work of Calderón-Garcidueñas and coworkers of quadruple aberrant protein pathology, including TDP-43 in children and young MMC adults [26].

**Core message**: Heavy traffic and indoor and outdoor PM_2.5_ exposures carry higher dementia risk, including Alzheimer’s and vascular dementia. PD risk and disease aggravation are linked to air pollution and ALS risk increases with Pb exposures and specific occupations (i.e., high exposures to diesel exhaust). AD, PD, and TDP-43 pathology start in childhood in the scenario of a polluted megacity.

## 7. Psychiatric Symptoms in Neurodegenerative Diseases

There is an overlap of neuro and psychiatric symptoms over the course of neurodegenerative diseases, including AD, PD, the behavioral variant of FTD (bvFTD), and ALS [52,53,54]. The reader is referred to the excellent review by Menculini’s group [53] describing a wide spectrum of psychopathology including mood, anxiety, and psychotic and obsessive compulsive symptoms associated with CNS disorders. It is also evident that apparently “functional” psychiatric disorders, such as depression and post-traumatic stress disorder, may represent precursors of dementia and neurodegeneration, and exposure to environmental pollutants may play a significant role in mediating this association [55,56,57,58]. Though associations between PM_2.5_ and a wide range of psychiatric disorders have been reported, the most consistent link has been demonstrated for mood disorders (particularly depression) and suicide [57,58]. Increases of 10 μg/m^3^ of chronic exposures to PM_2.5_ were associated with a ~18% increase in depression risk in Liu et al.’s [58] review and meta-analysis, where a subgroup analysis showed people aged ≥ 65 years from developed regions were at higher risk for depression. Young megacity residents are also at higher suicide risk if they are carriers of an apolipoprotein E4 allele [15]. In an autopsy study of 203 Metropolitan Mexico City MMC 25.36 ± 9.23-year-olds, APOE4 carriers had 4.92 times higher suicide risk vs. non-carriers having similar cumulative PM_2.5_ exposures and age, a situation very much relevant to the massive exposure in the US of millions of subjects during the 2020–2021 severe wildfire seasons [8].

Thus, in the developing ATX (N) classification for use across the AD continuum [59], we will need to include markers of exposure to air pollutants and cognitive instruments that will allow us to accurately explore the link between exposures and cognition impact in teens and young adults. Accurate documentation of smoking (tobacco and marijuana), industrial NPs exposures, and occupational and massive wildfire PM_2.5_ exposures needs to be done [8,11,12,38,42,45,47,49,60]. Environmental, industrial, and occupational PM exposures ought to be recorded for PD and ALS.

## 8. Conclusions

We have described the evolving landscape of PM pollution exposure effects and policy in the US. In the process we can identify some important lessons learned and the needs for future research. There is ample recognition that air pollution created mainly by fossil fuel combustion and other human activity adversely affects human health and the planet. While inhalation is an important and commonly recognized pathway for human exposure, particle ingestion and dermal absorption can be equally important. Similarly, indoor and outdoor air pollution contributes to the cumulative burden. Some viable behavioral and technological solutions to the PM emission problem have been identified and are beginning to be implemented. However, the transition to sustainability and a fossil fuel-free, cleaner environment will take time. Thus, we will need to continue to learn to deal with natural and human-made combustion emissions and other sources of particle pollution exposures. Specifically, exposures to traffic-related pollution, non-tailpipe vehicle emissions, engineered nanoparticles, and consumer product sources of particles will be around for some time. We cannot rest now. In fact, we should re-double all efforts at prevention and treatment of effects. This Review raises awareness specifically in the health community for these emerging concerns. Air quality practitioners around the country will continue working to bring all areas in the US into PM_2.5_ regional attainment. Meanwhile, the federal government is largely expected to follow the recommendations of the Independent Particulate Matter Review Panel [4] and adopt lower ambient PM air quality standards. However, the change and the improvements to follow will come about slowly. Additionally, the issues of UFP and NP emissions and exposures will most certainly remain unaddressed for reasons beyond this manuscript. Therefore, what should clinicians dealing with those sickened by PM pollution do?

They should request and get involved in federal-supported multidisciplinary epidemiological studies examining targeted neurological and psychiatric outcomes along accurate measurements and definitions of the chemical composition of PM_2.5_, UFPs and NPs. We have shown a neuropathology overlap, i.e., Aβ, P-tau, alpha-synuclein, and TDP-43 in young air pollution-exposed urbanites [15,26]. There is a common denominator for the quadruple aberrant protein pathology, are solid nanoparticles at the core of the problem?

Additionally, we need to educate health professionals not just about air pollution effects, but also the potential role that specific constituents of UFPs and NPs may play in inducing short and long term neurologic and psychiatric effects. While many additional toxic compounds besides the six criteria pollutants are included in management practices throughout the country under the federal Clean Air Act, many of these compounds can be present in the UFP or NP phase either as nuclei or adsorbed material; right now, exposure risks to these compounds are technically ignored. This notion brings to bare consideration of just how protective air quality policies that largely focus only on total PM mass truly are. Health professionals should be the loudest voice here. Additionally, physicians can play a major role in the education process: the relative contribution of indoor PM_2.5_ pollution (i.e., from common activities such as gas appliance use and smoking) is increasing and widely used engineered NPs (i.e., as in skin UV protectors) can result in high cumulative exposures. Clinicians can make patients aware of the adverse effects of exposure to PM pollution from wildfire smoke and other common sources. It may even be helpful to incorporate environmental exposure data to a medical history for potentially establishing a cumulative pollution dose profile. Exposures to complex mixtures of air pollutants and their health effects ought to be a greater public health priority, knowing that many of the higher risk neurological and psychiatric outcomes have fatal consequences and no current therapies have any impact on modification of their course.

We are all at risk and protecting the critical brain developmental periods should be a priority. Preventive medicine ought to be at work.

## Figures and Tables

**Figure 1 ijerph-18-11568-f001:**
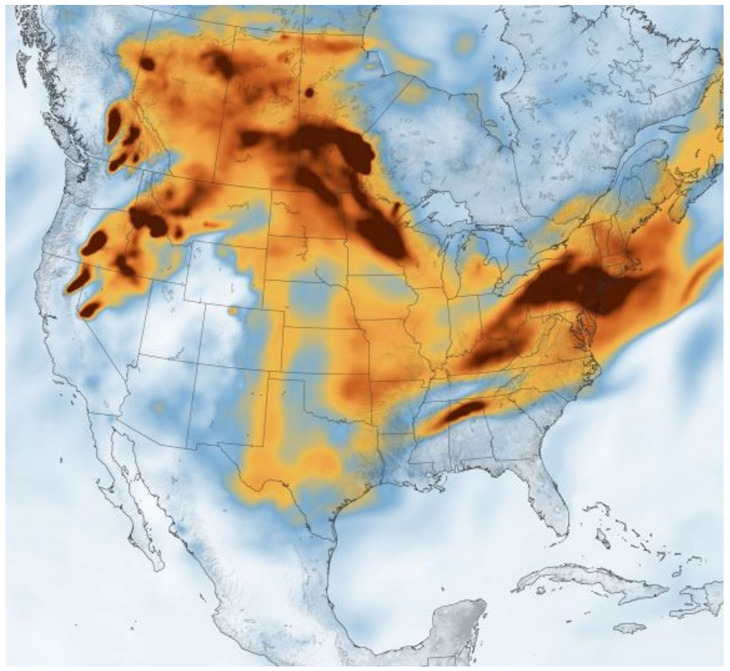
Black carbon particulates from wildfires spreads eastward across the U.S. 21 July 2021 (Image credit: Joshua Stevens/NASA Earth Observatory) Accessed on 24 July 2021.

**Figure 2 ijerph-18-11568-f002:**
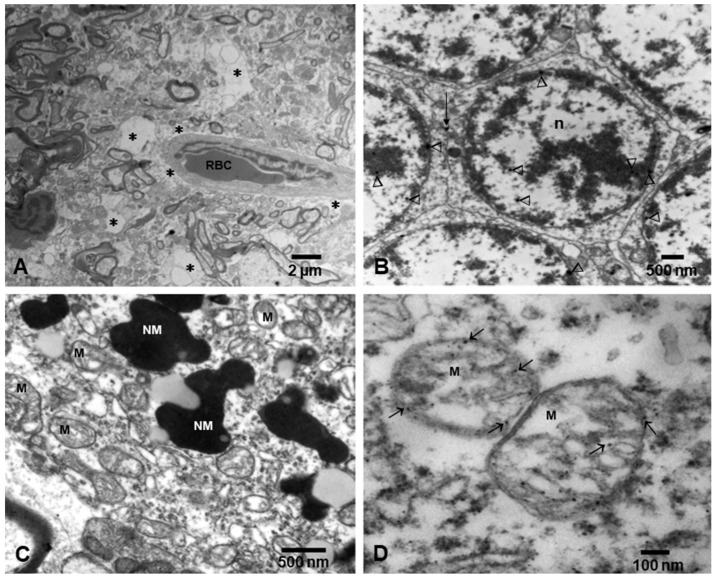
Electron micrographs of neurovascular unit (NVU) and neural organelles in Metropolitan Mexico City children. (**A**) Three-year-old boy, substantia nigrae pars compacta showing a capillary with one luminal red blood cell (RBC) surrounded by an extensively vacuolated, fragmented neuropil (*). (**B**) Cerebellar granular neurons from same child as (**A**), showing nanoparticles (arrowheads) in intranuclear location and at the membrane interphase between neurons (short arrow). (**C**) Fifteen-year-old substantia nigrae pars compacta showing numerous mitochondria (M) with abnormal cristae and neuromelanin structures (NM) with nanoparticles. (**D**) In a closeup, the mitochondria exhibit numerous nanoparticles in the matrix, cristae, and along the double layer mitochondrial wall (arrows).

**Figure 3 ijerph-18-11568-f003:**
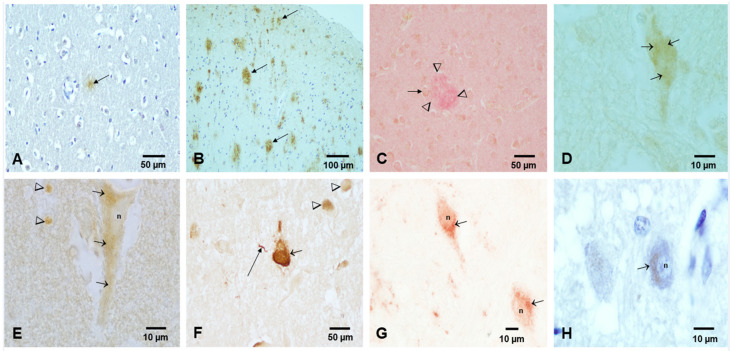
Light microscopy, immunohistochemistry (IHC) of aberrant neural proteins in MMC children and young adults. (**A**) Frontal cortex in an 11-month-old APOE 3/3 showing an Aβ diffuse plaque. IHC Aβ 17–24, 4G8, Covance, Emeryville, CA 1:1500, DAB brown product. (**B**) Temporal cortex, 11 y old child APOE 3/3 with multiple trans-cortical diffuse and mature Aβ plaques, 4G8 brown product. (**C**) Adult 36 y old, APOE 3/4, temporal cortex with mature amyloid plaques (arrowheads) and reactive astrocytes (short arrow). Double staining 4G8 red product, and glial acidic fibrillary protein (GAFP) brown product. (**D**) Two-year-old frontal cortex with granular cytoplasmic staining for hyperphosphorylated tau (P-tau) (short arrows). PHF-tau 8 phosphorylated at Ser 199–202-Thr 205, Innogenetics, Belgium, AT-8 1:1000, brown DAB product. (**E**) Eleven-year-old frontal cortex with pyramidal neuron granular cytoplasmic immunoreactivity for P-tau. (**F**) Forty-year-old male with numerous tangles supra and infratentorial, including substantia nigrae (short arrow), P-tau neurite (long arrow), and granular cytoplasmic immunoreactivity (arrowheads). AT-8 brown product. (**G**) Same 11 y old child as (**E**) with TPD-43 immunoreactivity in brainstem neurons, the nuclei (n) are negative, and the cytoplasm exhibits granular immunoreactivity (short arrows). TDP-43 red product. (**H**) Thirty-five-year old male, cochlear nuclei neurons positive for α-synuclein, phosphorylated at Ser-129, LB509, In Vitrogen, Carlsbad CA 1:1000. Red product counterstained with hematoxylin.

## Data Availability

Not applicable.

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
