# Peer review of "Particulate Air Pollution and Risk of Neuropsychiatric Outcomes. What We Breathe, Swallow, and Put on Our Skin Matters"

_ijerph, 2021, doi:10.3390/ijerph182111568_

Round 1

Reviewer 1 Report

This is an interesting review covering efficiently a wide range of aspects related to current evidence and concerns on the neuropsychiatric effects of air pollution along with other regulatory discussions. This is sound piece of work and I only have minor questions and comments:

The title seems a bit oversized and not very specific to the different topics covered in the manuscript.

Please define “US EPA” in the abstract. Also, all other abbreviations when used first should be introduced.

Also note that the WHO recently updated the AQGs, now the WHO-annual AQG for PM2.5 is 5 microgram/cubic metre, plus there are 4 Interim targets, WHO-IT4 is now 10. Please see https://apps.who.int/iris/handle/10665/345329. A further discussion concerning these new targets might be interesting for the reader.

Line 86: “This is Review is meant” – please correct.

When referring to wildfires it is important to note that  PM from wildfires causes more pronounced effects on mortality than urban PM, which is mostly due to smaller particle size (Verma V et al. Physicochemical

and toxicological profiles of particulate matter in Los Angeles during the October 2007 southern California wildfires. Environ Sci Technol 2009;43:954–960 AND Reid CE et al. Critical review of health impacts of wildfire smoke exposure. Environ Health Perspect 2016;124: 1334–1343.=

Is there any evidence concerning Swallowed NP/UFP and incidence of neuropsychiatric outcomes?

Linge 428: “importatn" please correct.

Author Response

REVIEWER #1 Comments and Authors Responses

This is an interesting review covering efficiently a wide range of aspects related to current evidence and concerns on the neuropsychiatric effects of air pollution along with other regulatory discussions. This is sound piece of work and I only have minor questions and comments:

The title seems a bit oversized and not very specific to the different topics covered in the manuscript.

AUTHORS RESPONSE: We thank this reviewer; the title now reads:

Particulate air pollution and risk of neuropsychiatric outcomes. What we breathe, swallow and put on our skin matters.

Please define “US EPA” in the abstract. Also, all other abbreviations when used first should be introduced.

AUTHORS RESPONSE: Done.

Also note that the WHO recently updated the AQGs, now the WHO-annual AQG for PM2.5 is 5 microgram/cubic metre, plus there are 4 Interim targets, WHO-IT4 is now 10. Please see https://apps.who.int/iris/handle/10665/345329. A further discussion concerning these new targets might be interesting for the reader.

AUTHORS RESPONSE: Thank very much. We have added a paragraph referring to the recent WHO guidelines (and the reference)

The 2021 WHO Global Air Quality Guidelines recommendation for annual PM2.5 Air Quality Guideline Level is 5µg/m3 , with four interim targets proposed as incremental steps in a progressive reduction of air pollution and intended for use in highly polluted areas [3].

[3] WHO global air quality guidelines. Particulate matter (PM2.5 and PM10), ozone, nitrogen dioxide, sulfur dioxide and carbon monoxide. Geneva: World Health Organization; 2021. Licence: CC BY-NC-SA 3.0 IGO.

Line 86: “This is Review is meant” – please correct.

AUTHORS RESPONSE: Thank very much. Corrected.

When referring to wildfires it is important to note that PM from wildfires causes more pronounced effects on mortality than urban PM, which is mostly due to smaller particle size (Verma V et al. Physicochemical

AUTHORS RESPONSE: Yes, we fully agreed with this reviewer, and the problem is only getting worse across the world.

Is there any evidence concerning Swallowed NP/UFP and incidence of neuropsychiatric outcomes?

AUTHORS RESPONSE: Good question. None to our knowledge. We have documented the small bowel NPs Disruption of epithelial integrity with TJ structural changes in MCMA v control dogs (p<0.0001), the major determinant of paracellular permeability characterized the MCMA dogs’ small bowel architecture. The Intestinal Barrier in Air Pollution-Associated Neural Invol vement in Mexico City Residents: Mind the Gut, the Evolution of a Changing Paradigm Relevant to Parkinson Disease Risk. Alzheimer’s Disease and Parkinsonism 2015

Linge 428: “importatn" please correct.

AUTHORS RESPONSE: Good catch. Thank you. Done.

Reviewer 2 Report

The review  of  Calderon-Garciduenas et al. “Particulate air pollution and higher risk of neurological and  psychiatric outcomes. What we breathe, swallow and put on our  skin matters” analyzed  the  correlations between air pollution , in particular PM /UFP and nervous system

The paper reviews this current major global public health issue but needs corrections before publication.

  1. Suggest removing from the title "psychiatric" because this aspect is mentioned only in the last chapter and for a few lines in the review. So unless the authors increase this part, the correct title is

Particulate air pollution and higher risk of neurological outcomes. What we breathe, swallow and put on our   skin matters

  1. Lanes 191-200 is not well written and not explains how UFP / NP can be involved in neuronal damage from the gut... I would recommend rewriting this part by adding supporting literature data (if any)

       3.Chapter 5 Stroke and air pollution is a bit confusing. I propose to the           authors to insert the IMPORTANT and summary sentence of the damage from UFP  (…Inflammation, oxidative stress and endothelial dysfunction are three key underlying molecular mechanisms associated to ambient air pollution exposures and targeted cerebrovascular and  neuropsychiatric disorders [30]. Lanes 330-332 ) in chapter 4 before ... dysfunctional mitochondria are ... lane 219 and chapter 5 incorporate it in chapter 4

  1. In chapter 4, the inflammatory action of the UFP / NP against the glia is never mentioned. I suggest the authors insert text / references regarding this point
  2. Attention to font size (lines 291-301; 394-398; 361-366)
  3. References, remove the underline in some references (24 25 26 50)

Author Response

REVIEWER #2 Comments and Authors Responses

The review of Calderon-Garciduenas et al. “Particulate air pollution and higher risk of neurological and psychiatric outcomes. What we breathe, swallow and put on our  skin matters” analyzed  the  correlations between air pollution , in particular PM /UFP and nervous system

The paper reviews this current major global public health issue but needs corrections before publication.

  1. Suggest removing from the title "psychiatric" because this aspect is mentioned only in the last chapter and for a few lines in the review. So unless the authors increase this part, the correct title is

AUTHORS RESPONSE:  We thank this reviewer, the title now reads:

Particulate air pollution and risk of neuropsychiatric outcomes. What we breathe, swallow and put on our skin matters.

  1. Lanes 191-200 is not well written and not explains how UFP / NP can be involved in neuronal damage from the gut... I would recommend rewriting this part by adding supporting literature data (if any)

AUTHORS RESPONSE: Thank you very much. We have added a full paragraph to the text:

Swallowed NP/UFP has an easy access to the GI epithelium and submucosa and their damage allows direct access to the enteric nervous system [17]. In a combined study of gastric, small bowel and vagal nerves at the cervical level in children, young adults and dogs with low versus high exposures to PM 2.5, NPs were abundant in erythrocytes, unmyelinated submucosal, perivascular and intramuscular nerve fibers, ganglionic neurons and vagal nerves and associated with organelle pathology, in highly exposed urbanites[17]. Immunohistochemistry showed hallmarks of Parkinson and Alzheimer’s diseases including aggregated alpha-synuclein and hyperphosphorylated tau in gastrointestinal tract and vagal nerves of young children and Mexico City young adults [17].

  1. Chapter 5 Stroke and air pollution is a bit confusing. I propose to the authors to insert the IMPORTANT and summary sentence of the damage from UFP (…Inflammation, oxidative stress and endothelial dysfunction are three key underlying molecular mechanisms associated to ambient air pollution exposures and targeted cerebrovascular and  neuropsychiatric disorders [30]. Lanes 330-332 ) in chapter 4 before ... dysfunctional mitochondria are ... lane 219 and chapter 5 incorporate it in chapter 4

AUTHORS RESPONSE: To make the paragraph clear, we changed the statement to read as follows:

Factors such as inflammation, oxidative stress and endothelial dysfunction associated to ambient air pollution exposures are targeting cerebrovascular and neuropsychiatric disorders [30].

  1. In chapter 4, the inflammatory action of the UFP / NP against the glia is never mentioned. I suggest the authors insert text / references regarding this point

AUTHORS RESPONSE: Yes, indeed glia are key in the response to air pollutants. We are very familiar with some of the recent literature and the topic is so important that truly deserves a paper on their own.

Rodriguez-Campuzano et al., Acute Exposure to SiO 2 Nanoparticles Affects Protein Synthesis in Bergmann Glia Cells. Neurotox Res doi: 10.1007/s12640-019-00084-0

Attention to font size (lines 291-301; 394-398; 361-366)

AUTHORS RESPONSE: Yes.

References, remove the underline in some references (24 25 26 50)

AUTHORS RESPONSE: Done.